# Noncovalent interaction with a spirobipyridine ligand enables efficient iridium-catalyzed C–H activation

Yushu Jin [1], Boobalan Ramadoss[1], Sobi Asako [1] ✉ & Laurean Ilies [1] ✉

Exploitation of noncovalent interactions for recognition of an organic substrate has received much attention for the design of metal catalysts in organic synthesis. The CH–π interaction is especially of interest for molecular recognition because both the C–H bonds and the π electrons are fundamental properties of organic molecules. However, because of their weak nature, these interactions have been less utilized for the control of organic reactions. We show here that the CH–π interaction can be used to kinetically accelerate catalytic C–H activation of arenes by directly recognizing the π-electrons of the arene substrates with a spirobipyridine ligand. Computation and a ligand kinetic isotope effect study provide evidence for the CH–π interaction between the ligand backbone and the arene substrate. The rational exploitation of weak noncovalent interactions between the ligand and the substrate will open new avenues for ligand design in catalysis.

Noncovalent interactions are prevalent in Nature, and have been extensively used by chemists for molecular recognition, building complex hierarchical structures, and controlling reactivity and selectivity in the fields of crystal engineering, supramolecular chemistry, organic synthesis, and catalysis[1–7] Among noncovalent interactions, the CH–π interaction is one of the weakest; nevertheless, because both the C–H bonds and the π electrons are fundamental properties of organic molecules, manipulation of these interactions could provide a general strategy for molecular and reactivity control[8–17] Because of their weak nature, while CH–π interactions have been used to thermodynamically stabilize a molecular system, sometimes synergistically, they have been less exploited for stabilizing a transition state in catalysis[3] and evidence for their involvement is scarce and largely limited to computational studies. A notable early example is a report by Noyori on an enantioselective transfer hydrogenation of aryl ketones catalyzed by chiral η⁶-ruthenium complexes, where a CH–π interaction between the C–H bond of the ligand and the aryl substituent of the ketone was proposed by computational studies to stabilize one of the diastereomeric transition states[9]. Several subsequent studies also proposed through computation the stabilization of a transition state by CH–π interaction as a rationale for the observed enantio-[10–14] and regioselectivity[15,16], including in

transition-metal-catalyzed C–H functionalization (Fig. 1a). While stabilization by interaction between the C–H bond of the substrate and the π electrons of the ligand is more common (Fig. 1a, left), Musaev and coworkers proposed by computation an interaction between the C–H bond of the ligand and the π electrons of the incoming arene substrate in palladium-catalyzed directed C–H activation (Fig. 1a, right)[10]. During our studies on the development of ligands for molecular recognition in transition-metal-catalyzed C–H functionalization[18–20], we envisioned that the CH–π interaction could be used to recognize the π-electrons[3,21–24], an inherent property of all arenes, and thereby stabilizing the key transition state of the undirected aromatic C–H bond cleavage step by rational ligand design (Fig. 1a, right). Here we show that the CH–π interaction can indeed kinetically accelerate the iridium-catalyzed undirected C–H borylation of arenes (Fig. 1c), by directly recognizing the π-electrons of the arene substrate through a spirobipyridine ligand. Computation and a ligand kinetic isotope effect (KIE) study provided evidence for the involvement of the CH–π interaction between the C–H bond of the ligand backbone and the π-electrons of the arene substrate.

Transition-metal-catalyzed borylation of a C–H bond in an arene[25–31] is a straightforward method for the preparation of aromatic organoboron compounds[32], widely used as substrates for

¹RIKEN Center for Sustainable Resource Science, Wako, Saitama, Japan. ✉e-mail: sobi.asako@riken.jp; laurean.ilies@riken.jp

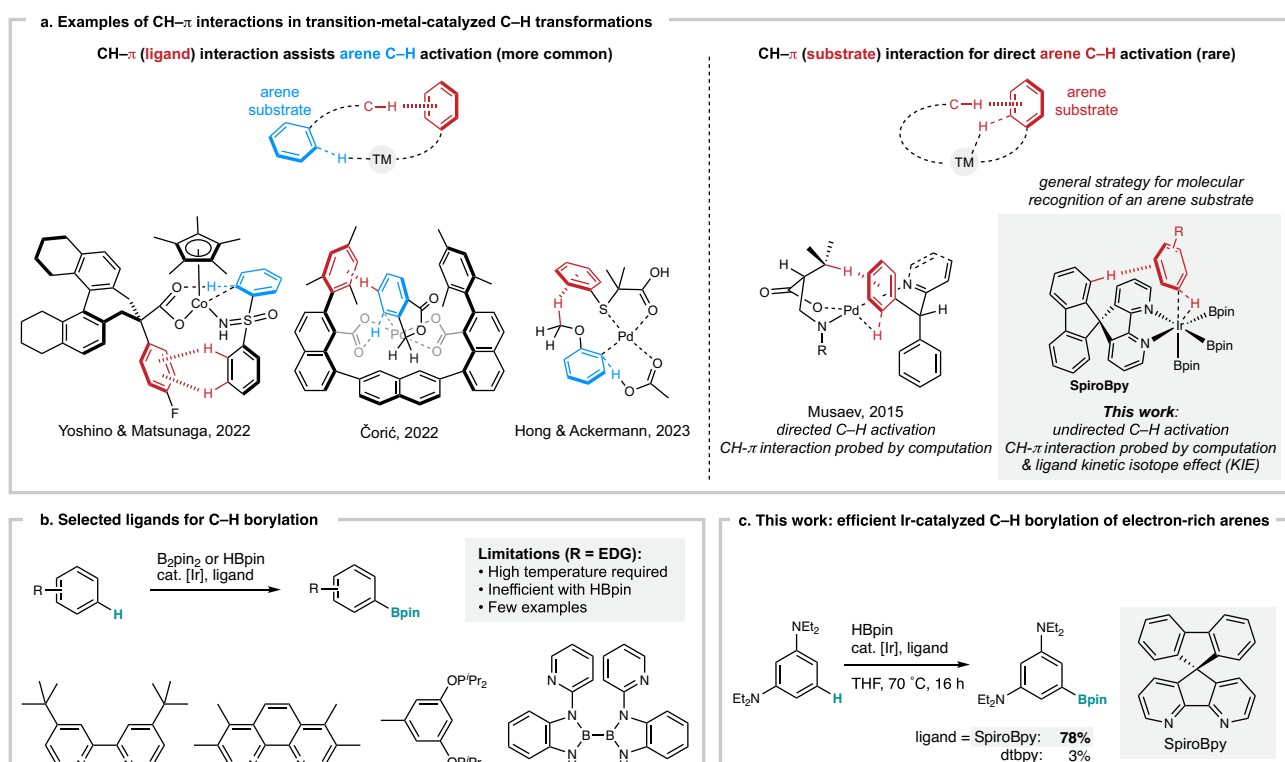

**Fig. 1 | CH–π interactions in transition metal catalysis. a** Examples of CH–π interactions proposed computationally to stabilize a transition state in transition metal catalysis. **b** Selected active ligands for the C–H borylation of arenes. **c** Efficient C–H borylation of electron-rich arenes enabled by spirobipyridine (SpiroBpy) ligand through CH–π interaction. TM transition metal. pin pinacolato. *i*Pr isopropyl. Et ethyl. EDG electron-donating group. THF tetrahydrofuran.

Suzuki–Miyaura cross coupling. Extensive investigations to date have identified bidentate nitrogen compounds such as dtbpy[33], tmphen[34,35], and other compounds[36–47] as highly efficient ligands for iridium-catalyzed C–H borylation (Fig. 1b). However, the reaction of a stoichiometric amount of electron-rich substrate under mild conditions remains challenging, especially when pinacolborane (HBpin) is used as the borylation reagent[34]. Our group is interested in the design of catalysts that can recognize an organic substrate through non-covalent interactions to control reactivity and selectivity[19,20,48–53]. Whereas tuning of the electronic and steric properties for commonly used bipyridine ligands such as dtbpy or tmphen is achieved by introducing substituents into the plane of the molecule, we envisioned that placing a fluorene moiety perpendicularly on the bipyridine core (i.e., SpiroBpy[20] in Fig. 1a, c) would enable an attractive non-covalent interaction, such as CH–π interaction with an arene substrate, resulting in acceleration of iridium-catalyzed C–H borylation. Herein, we report that a spirobipyridine ligand (SpiroBpy) surpasses the efficiency of the ligands used to date and enabled iridium-catalyzed borylation of various electron-rich arenes with HBpin in high yields at 50–70 °C.

## Results and discussion
### Optimization of reaction conditions
We started our examination with the iridium-catalyzed borylation of 1,3-dimethoxybenzene (**1a**) with HBpin, which typically shows lower reactivity than that of B$_2$pin$_2$ (Table 1)[34]. With the commonly used ligand dtbpy, the borylated product **2a** was obtained in low yield (20%, entry 1). 3,4,7,8-Tetramethyl-1,10-phenanthroline (tmphen), reported as one of the most efficient ligands for iridium-catalyzed borylation[34,35], gave a modest yield (50%, entry 2). We next examined the borylation of **1a** using several substituted R-SpiroBpy ligands (entries 3–6), and we

found that they gave yields up to 82%. The substituents on the fluorene backbone of SpiroBpy affected the reactivity, and the simplest, pristine SpiroBpy (R = H) proved the best ligand, affording the borylated product **2a** in 82% yield (78% after isolation, entry 3). This result is consistent with our hypothesis that a stronger CH–π interaction would be observed with the less bulky ligand. The reaction was clean, and we observed only 13% recovery of the starting material **1a**. The importance of the spirobipyridine motif for accelerating the reaction is illustrated by the modest performance of the methylene-bridged bipyridine ligand **L1**[54] (entry 7), despite the bite angles of **L1** and SpiroBpy being very similar. Introducing two phenyl groups into the methylene bridge (**L2**) resulted in lower 65% yield (entry 8). This result suggests that a certain degree of noncovalent interaction is involved, but not as significant as for the rigid SpiroBpy ligand. The reaction proceeded slower at 30 °C (entry 9). An essentially stoichiometric amount of HBpin (120 mol%) gave a slightly lower yield (entry 10). Other organic solvents such as cyclohexane could also be used (entry 11). For a detailed investigation of the reaction parameters, see Supplementary Table 1.

To probe further the acceleration effect of the SpiroBpy ligand, we monitored the progress of the iridium-catalyzed C–H borylation of electron-rich substrates (**1a** and **1m**) using SpiroBpy and tmphen under the optimized conditions (Supplementary Figs. 1 and 2), to find that the borylation using SpiroBpy consistently proceeds faster than that of tmphen, especially at the initial stages of the reaction.

### Scope of substrates
The SpiroBpy ligand proved highly efficient for the iridium-catalyzed borylation of a variety of arenes, as compared with the dtbpy or tmphen ligands (Fig. 2). We focused our investigation of the borylation on electron-rich arenes, considered challenging substrates for this

## Table 1 | Effect of ligands and key reaction parameters

| Entry | Ligand | 2a (%) | 1a (%) |
|---|---|---|---|
| 1 | dtbpy | 20 | 78 |
| 2 | tmphen | 50 | 44 |
| 3 | SpiroBpy | 82 (78)[a] | 13 |
| 4 | Bpin-SpiroBpy | 48 | 47 |
| 5 | Ph-SpiroBpy | 61 | 31 |
| 6 | tBu-SpiroBpy | 80 | 8 |
| 7 | **L1** | 26 | 69 |
| 8 | **L2** | 65 | 23 |
| 9[b] | SpiroBpy | 34 | 59 |
| 10[c] | SpiroBpy | 56 | 38 |
| 11[d] | SpiroBpy | 79 | 9 |

Reaction conditions: **1a** (0.10 mmol), HBpin (200 mol%), [Ir(OMe)(cod)]$_2$ (2 mol%), ligand (4 mol%), THF (1.0 mL), 16 h at 50 °C. The yield was determined using GC in the presence of hexadecane as an internal standard, after calibration. [a]Yield of the isolated product in parentheses. [b]At 30 °C. [c]With HBpin (120 mol%). [d]Cyclohexane as the solvent. cod 1,5-cyclooctadiene. Ph phenyl. tBu tertiary butyl.

reaction[27–30]. For convenience in studying the efficiency of the reaction without interference from regioselectivity issues, we chose *meta*- or *ortho*-disubstituted, and polysubstituted arenes as the substrate. Since the C–H bond in our SpiroBpy ligand can recognize the π system, a fundamental property of all arenes, the CH–π acceleration strategy is applicable to a wide range of aromatic substrates. The SpiroBpy ligand gave high yields for arenes possessing multiple electron-donating groups such as methoxy, amino, alkyl, or silyl; for these electron-rich substrates, dtbpy gave mostly low yields, and tmphen was also consistently less effective, especially for aniline and alkylbenzene derivatives. Borylation of anisole derivatives **1a−d** proceeded in high yield with SpiroBpy (>80% yield), whereas the reaction using tmphen proceeded with a lower yield, especially for **1a** and **1c**. Triisopropylsilyl (TIPS)-protected phenol **1e** also reacted well with SpiroBpy, while both dtbpy and tmphen gave low yields. *meta*-Terphenyl derivative **1f** could be borylated in quantitative yield using SpiroBpy. SpiroBpy showed high reactivity with alkylbenzenes[55], affording products **2g−j** in high yield (72−96%), whereas tmphen and dtbpy gave low yields (<30%). Aniline derivatives are compounds of great importance for all areas of chemistry, but these electron-rich arenes are difficult substrates for iridium-catalyzed borylation[27–30]. We found that SpiroBpy is also a highly efficient ligand for the borylation of diaminobenzene derivatives (**1l−q, 1v, 1w**) at 70 °C. Notably, in the case of diaminobenzenes **1m−p**, the borylation proceeds in very low yield in the presence of tmphen or dtbpy, whereas SpiroBpy gave the corresponding borylated products in high yield. More reactive substrates (**1r−x**) possessing a halogen group and a methoxy, alkyl, silyl, or amino group reacted in high yields in the presence of the SpiroBpy ligand; for these reactive substrates, dtbpy and tmphen ligands also performed well, albeit with consistently lower yields than SpiroBpy. Thus, the SpiroBpy ligand is a generally effective ligand for the iridium-catalyzed borylation of a variety of arenes with HBpin.

We also demonstrated the gram-scale borylation of arenes[56] of interest for medicinal chemistry. Lidocaine (**3a**), a local anesthetic, was borylated in 62% yield, and a phenylalanine derivative (**3b**) was borylated in 96% yield[57]; the synthetically versatile boroester group enables access to new chemical space, of importance for drug discovery. Additionally, the borylation of 1 g of indole derivative **3c** with our

SpiroBpy ligand to give **4c** in 95% yield, an intermediate in the synthesis of tambromycin, a natural product[58,59].

## Mechanistic investigations

We performed a computational investigation to reveal the reasons behind the high activity of the SpiroBpy ligand (Fig. 3). Thus, we used 1,3-bis(dimethylamino)benzene (**5**) as a model substrate, and we compared **L**IrBpin$_3$ complexes bearing **L1**, SpiroBpy, and tmphen. As expected, we found that (SpiroBpy)IrBpin$_3$ has the lowest barrier for the C–H cleavage step (**TS$_{BC}$**: 32.8 kcal mol$^{-1}$ (**L1**), 30.3 kcal mol$^{-1}$ (SpiroBpy), 32.1 kcal mol$^{-1}$ (tmphen)) (Fig. 3a). A distortion/interaction analysis[60,61] indicated that the decrease in activation energy is mainly attributed to an increase in interaction energy in the transition state of the (SpiroBpy)IrBpin$_3$ complex ($\Delta E^{\ddagger}_{int}$: −53.2 kcal mol$^{-1}$ (**L1**), −56.0 kcal mol$^{-1}$ (SpiroBpy), −53.2 kcal mol$^{-1}$ (tmphen)). We conjecture that a noncovalent interaction (NCI) between the C–H bond of the ligand backbone, which is only present in three-dimensionally expanded SpiroBpy, and the π-electrons of the arene substrate is a key interaction responsible for the rate enhancement. This was further corroborated by an independent gradient model based on the Hirshfeld partition (IGMH) analysis (Fig. 3b)[62–64] and an NCI plot analysis (Supplementary Fig. 6)[62–64], which showed an attractive interaction (green) between the C–H bond of the SpiroBpy backbone and the arene, both in the transition state and in the resulting Ir(V) complex **C**. An NBO analysis also indicated that donor−acceptor interactions between them (donor, π orbitals of the arene substrate; acceptor, the C–H σ* orbital of the ligand backbone) could stabilize **TS$_{BC}$** and **C**, which were found to be larger for the more electron-rich substrate (Supplementary Table 2). This is consistent with the experimentally observed strong acceleration effect by SpiroBpy when electron-rich diaminobenzene substrates were used. While this attractive noncovalent interaction alone may not fully account for the calculated stabilization of the TS, we propose that it plays a major role.

To gain experimental evidence for the involvement of a CH–π interaction, we next synthesized an octadeuterated ligand (SpiroBpy-$d_8$), and studied the effect of replacing H on the ligand backbone with D on the reaction rate. As shown in Fig. 4, the reaction of 1,3-

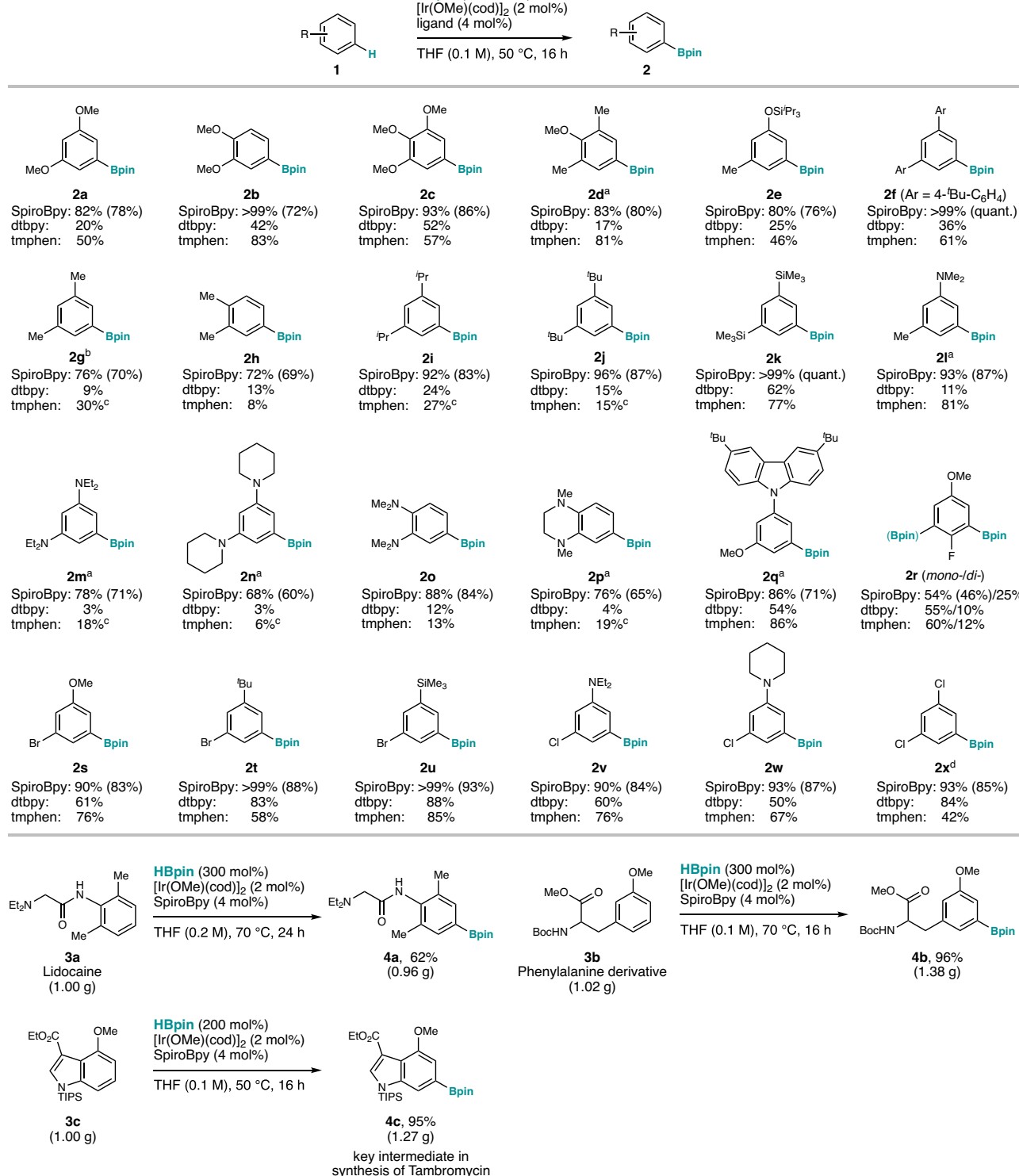

**Fig. 2 | Scope of substrates.** Reaction conditions: **1** (0.10 mmol), HBpin (200 mol%), [Ir(OMe)(cod)]₂ (2 mol%), ligand (4 mol%), THF (1.0 mL), 50 °C, 16 h. Yields were determined using GC (after calibration) or ¹H NMR in the presence of an internal standard. Yields of the isolated products are shown in parentheses. See the Supplementary Information for details. Iridium-catalyzed C–H borylation of pharmaceutically relevant compounds on a gram scale is shown at the bottom. ᵃAt 70 °C. ᵇAt 60 °C. ᶜAverage of two runs. ᵈAt room temperature. Me methyl. Boc *tert*-butyloxycarbonyl. TIPS triisopropylsilyl.

bis(dimethylamino)benzene (**5**) with HBpin in the presence of the iridium catalyst and SpiroBpy or SpiroBpy-$d_8$ was conducted in parallel at three different temperatures (two times each), to reveal an inverse kinetic isotope effect (KIE) (i.e., the reaction is faster when using SpiroBpy-$d_8$). This supports the involvement of an interaction between the C–H bond of the ligand backbone and the arene substrate in the transition state of the turnover-limiting step. While a detailed discussion is premature at this stage, based on a differential Eyring analysis (Supplementary Fig. 5), we found that the reaction using SpiroBpy is enthalpically favored, while the one using SpiroBpy-$d_8$ is entropically favored, in agreement with the stronger KIE at higher temperature and previous reports[65,66]. Although speculative at the moment, this could be

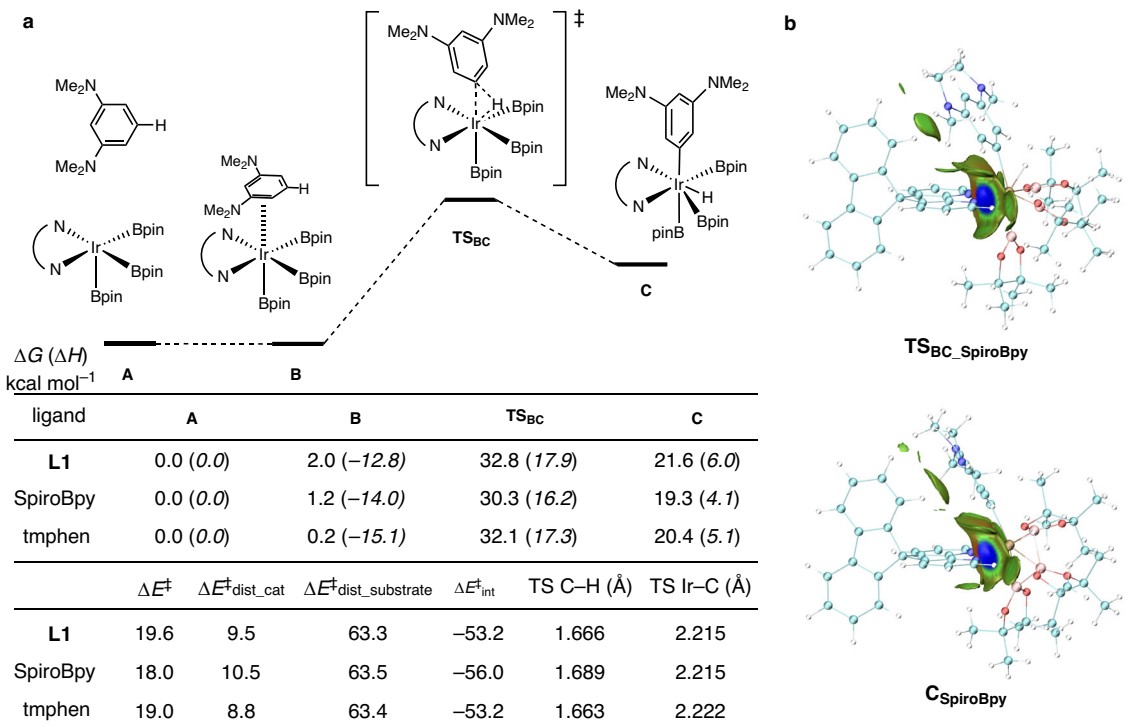

**Fig. 3 | Computational investigations. a** Relative Gibbs energies and enthalpies (*italicized*) calculated at the M06/SDD:6–311+G(d,p)$_{THF(SMD)}$//B3LYP-D3/SDD:6–31+G(d,p) level of theory (298.15 K) are shown for the C–H cleavage of 1,3-(Me$_2$N)$_2$benzene with **L**IrBpin$_3$. **A** = Separated **L**IrBpin$_3$ and 1,3-(Me$_2$N)$_2$benzene, **B** = π-arene complex, **TS$_{BC}$** = C–H oxidative addition transition state, **C** = **L**Ir(V)

ArHBpin$_3$. **b** Independent gradient model based on the Hirshfeld partition (IGMH) analysis for **TS$_{BC\_SpiroBpy}$** and **C$_{SpiroBpy}$** (fragment 1, SpiroBpy; fragment 2, IrBpin$_3$_arene), mapped with sign($\lambda_2$)$\rho$ colored isosurfaces of $\delta g^{inter}$ = 0.004 a.u. Color code: cyan, carbon; white, hydrogen; pink, boron; blue, nitrogen; red, oxygen; ochre, iridium. TS transition state.

| ligand | A | B | TS$_{BC}$ | C |
|---|---|---|---|---|
| **L1** | 0.0 (*0.0*) | 2.0 (*−12.8*) | 32.8 (*17.9*) | 21.6 (*6.0*) |
| SpiroBpy | 0.0 (*0.0*) | 1.2 (*−14.0*) | 30.3 (*16.2*) | 19.3 (*4.1*) |
| tmphen | 0.0 (*0.0*) | 0.2 (*−15.1*) | 32.1 (*17.3*) | 20.4 (*5.1*) |

| | $\Delta E^{\ddagger}$ | $\Delta E^{\ddagger}_{dist\_cat}$ | $\Delta E^{\ddagger}_{dist\_substrate}$ | $\Delta E^{\ddagger}_{int}$ | TS C–H (Å) | TS Ir–C (Å) |
|---|---|---|---|---|---|---|
| **L1** | 19.6 | 9.5 | 63.3 | −53.2 | 1.666 | 2.215 |
| SpiroBpy | 18.0 | 10.5 | 63.5 | −56.0 | 1.689 | 2.215 |
| tmphen | 19.0 | 8.8 | 63.4 | −53.2 | 1.663 | 2.222 |

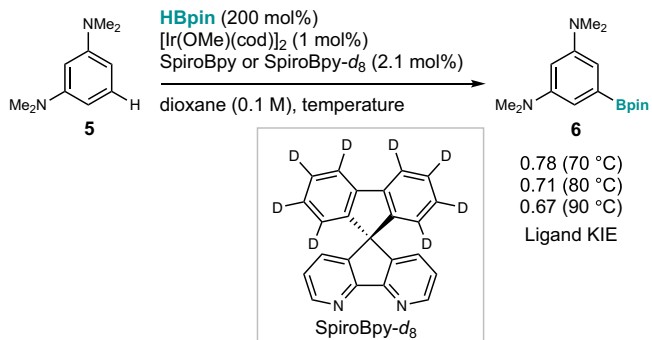

**Fig. 4 | Mechanistic investigation.** Ligand kinetic isotope effect (KIE).

partially explained by the stronger CH–π interaction and shorter CH–π distance, which are a result of the longer C–H bond than the C–D bond and the slightly larger polarizability of hydrogen than that of deuterium.

In summary, we found that SpiroBpy is an efficient ligand for iridium-catalyzed C–H borylation of arenes with HBpin, including electron-rich arenes possessing multiple alkoxy, amino, alkyl, or silyl groups, which react poorly with other bipyridine ligands. A mechanistic study suggested that the increase in reactivity may be ascribed to an attractive interaction between the C–H bond of the spirobipyridine backbone and the π-electrons of the arene. Because π-electrons are an innate property of all arenes, the acceleration strategy based on CH–π interaction is expected to be general for the C–H functionalization of aromatic substrates. This is in contrast to strategies based on other noncovalent interactions such as hydrogen bonding, Lewis acid–base, and ion-pair interactions, which require a specific substituent or heteroatom on the arene, resulting in an inherent limitation in scope. We expect that spirobipyridine derivatives will find broad applications as a

ligand in transition metal catalysis[67], and we are further investigating the use of attractive interactions to accelerate catalytic C–H activation. We also believe that ligand KIE studies, largely ignored to date[42], are a useful tool for investigating noncovalent interactions between the catalyst and substrate, and we are working towards better understanding of these effects.

## Methods

### General procedure for the iridium-catalyzed C–H borylation of electron-rich arenes with SpiroBpy ligand

An oven-dried J-young Schlenk tube (ca. 13 mL) fitted with a septum was charged with [Ir(OMe)(cod)]$_2$ (1.3 mg, 2 μmol, 2 mol%) and SpiroBpy (1.3 mg, 4 μmol, 4 mol%), then it was evacuated and purged with nitrogen gas three times. When the arene substrate **1** (0.10 mmol) was a solid, it was also added together with [Ir(OMe)(cod)]$_2$ and SpiroBpy. When the arene substrate **1** (0.10 mmol) was oil, it was added via syringe under a nitrogen atmosphere. Next, dry THF (1.0 mL) and pinacolborane (25.6 mg, 0.20 mmol) were added via (micro)syringe and the reaction mixture was stirred at 50 °C for 16 h. Upon heating, the reaction mixture turned dark brown and appeared homogeneous. After cooling to room temperature, the reaction mixture was diluted with EtOAc. The yield of **2** was determined by analyzing the crude mixture by GC (with hexadecane as an internal standard) or $^1$H NMR (with 1,3,5-trimethoxybenzene as an internal standard). After removing all volatiles under reduced pressure, the product was isolated by silica gel column chromatography or gel permeation chromatography (GPC).

### Representative example: iridium-catalyzed C–H borylation of $N^1,N^1,N^3,N^3$-tetraethylbenzene-1,3-diamine (1m) with SpiroBpy ligand

An oven-dried J-young Schlenk tube (ca. 13 mL) fitted with a septum was charged with [Ir(OMe)(cod)]$_2$ (1.3 mg, 2 μmol, 2 mol%) and

SpiroBpy (1.3 mg, 4 μmol, 4 mol%), then it was evacuated and purged with nitrogen gas three times. $N^1,N^1,N^3,N^3$-tetraethylbenzene-1,3-diamine **1m** (22.0 mg, 0.10 mmol) was added via syringe under a nitrogen atmosphere. Next, dry THF (1.0 mL) and pinacolborane (25.6 mg, 0.20 mmol) were added via (micro)syringe and the reaction mixture was stirred at 70 °C for 16 h. After cooling to room temperature, the reaction mixture was diluted with EtOAc. The yield of product **2m** was determined by $^1$H NMR analysis of the crude mixture using 1,3,5-trimethoxybenzene as an internal standard to be 78% yield. The crude mixture was purified by GPC (eluent: CHCl$_3$) to afford the target compound **2m** as a colorless solid (24.6 mg, 0.071 mmol, 71%).

## Data availability

The data supporting the findings of this study are available within the paper and its Supplementary Information, or from the authors upon request. Detailed conditions for each reaction, compound characterization data, kinetic experiment data, and computational data are provided in the Supplementary Methods, Supplementary Figs. 1–7, Supplementary Tables 1–3, and the Source Data file. NMR spectra are available in Supplementary Figs. 8–97. Source data are provided with this paper.

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

## Acknowledgements

This work was supported by the Japan Society for the Promotion of Science (JSPS) KAKENHI Grant-in-Aid for Transformative Research Areas No. JP22H05384 (Digi-TOS) (L.I.), Grant-in-Aid for Scientific Research (B) No. JP22H02125 (S.A.), Grant-in-Aid for Early-Career Scientists No. JP22K14689 (Y.J.), Basic Science Research Grant from the Sumitomo Foundation (L.I.), Shorai Foundation for Science and Technology (S.A.), and RIKEN Incentive Research Project grant (S.A.). Computational time provided by the Research Center for Computational Science, Okazaki, Japan (Projects: 23-IMS-C064, 22-IMS-C072, 21-IMS-C072) and the RIKEN HOKUSAI BigWaterfall are gratefully acknowledged. We thank Dr. Zhaomin Hou and Dr. Masanori Takimoto (RIKEN) for generously allowing us to use the mass spectrometer, and Dr. Shuichi Hiraoka (The University of Tokyo), Dr. Kazuhiro Okamoto (RIKEN), Dr. Yoshihiro Sohtome (RIKEN), Dr. Hideshi Ooka (RIKEN), Dr. Tatsuhiko Yoshino (Hokkaido University), and Dr. Shuhei Kusumoto (The University of Tokyo) for helpful discussions.

## Author contributions

Y.J., S.A., and L.I. designed the experiments, B.R. performed preliminary experiments, and Y.J. performed the experiments. S.A. performed the computational studies. S.A. and L.I. directed the research. Y.J., S.A., and L.I. wrote the manuscript. All authors analyzed the data and contributed to discussions.

## Competing interests

The authors declare no competing interests.

## Additional information

Sobi Asako or Laurean Ilies.

**Peer review information** *Nature Communications* thanks Tian Lu and the
other anonymous reviewer(s) for their contribution to the peer review of
this work. A peer review file is available.

