## [Peer Review File · Nature Communications]

Noncovalent Interaction with a Spirobipyridine Ligand Enables Efficient Iridium-Catalyzed C-H ActivationREVIEWER COMMENTS

Reviewer #1 (Remarks to the Author):

Comments: In this study, the authors present an excellent strategy for the iridium-catalyzed C–H borylation of electron-rich di-substituted arenes, employing a nonconventional spirobipyridine ligand. This ligand, akin to their previously utilized ligand for undirected meta-selective C–H activation of arenes through remote steric control, exhibits significantly enhanced reactivity compared to conventionally used dtbpy and tmphen ligands. The authors attribute this heightened reactivity to the involvement of a CH– π interaction between the C–H bond of the ligand backbone and the π -electrons of the arene substrate, as proposed through computational studies and ligand kinetic isotope effect analysis. The comprehensive substrate scope exploration includes electron-rich di-substituted arenes, demonstrating high isolated yields and broad functional group tolerance. Additionally, the borylation of few medicinally relevant arenes is showcased.

The authors' assertion about the pivotal role of CH– π interaction in the increased reactivity of the SpiroBpy ligand is intriguing. I share the authors' vision that the thoughtful exploration of such weak noncovalent interactions between ligands and substrates holds promise for advancing ligand design in transition metal catalysis.

While the work is compelling, I have some queries and suggestions that I believe could enhance the clarity and impact of the study. Following necessary revisions, this work has the potential to make a valuable contribution to Nature Communications and capture the attention of the synthetic chemistry community.

1. It is requested to the authors to take into account the acidity of the ortho proton in the ligand participating in the CH– π interaction with the substrate. To improve conformation, it may be useful to introduce an electron-withdrawing substitution adjacent to that proton, rather than Ph or Bpin, to increase the acidity of the proton and enhance its participation in the CH– π interaction.
2. In the introduction part an important review for the use of noncovalent interaction is missing (Tetrahedron Letters 59, 1269-1277). It is suggested to cite this reference.
3. To gain better insights into whether the π electron cloud of the substrate is indeed involved in non-covalent interaction with the ligand's ortho proton, it would be nice to check a ligand with a specific substitution (e.g., methyl group) at that position. This alteration aims to provide clearer clarification.
4. In the KH/KD experiments, the authors replaced all protons in the ligand with deuterium, creating uncertainty about which specific proton of the ligand is engaged in the proposed non-covalent interaction. In my opinion, the authors might enhance clarity by introducing deuterium solely at the ortho position of the ligand and conducting the control experiment accordingly.
5. Given the significance of the π cloud in substrate interactions, the authors opted for di-substituted arenes with electron-donating properties. However, for a comprehensive comparison and to validate the proposed hypothesis, it would be nice if the authors may examine the reaction with electron-deficient arenes.
6. If possible, it would be beneficial to present examples involving mono-substituted arenes, encompassing both electron-rich and electron-deficient substrates such as anisole, N,N-dimethyl aniline, tBu-benzene, trifluoromethyl benzene, among others.
7. Finally, it is suggested that the authors include citations to other pertinent research papers and review articles addressing non-covalent interactions and catalyst engineering. (Such as: Chem. Soc. Rev., 51, 5042–5100 (2022), J. Am. Chem. Soc., 143, 5022–5037 (2021),

Science Advances 2023, 9, eadg3311.).

Collectively, I have enjoyed a lot to read this paper and I strongly believe that this contribution would certainly be impactful for the catalysis research and for all types of synthetic organic chemistries.

Reviewer #2 (Remarks to the Author):

The author proposed a novel method to improve the efficiency of Ir complex-catalyzed C-H borylation of arenes, that is, through the appropriate selection of Ir ligands, stabilization of transition state of the undirected aromatic C-H bond cleavage step is achieved by forming C-H...pi non-covalent interaction between the ligand and the substrate. The author proved the effectiveness of this idea through comprehensive experiments and screened out the most ideal ligand, spirobipyridine. The article further demonstrates from a theoretical perspective that C-H...pi interaction improves catalytic efficiency through rigorous DFT calculations, and clearly demonstrates the existence of C-H...pi interaction through the very intuitive IGMH analysis. This is an outstanding work that combines theory and experiment, and it is also a very successful example of utilization of weak interactions to achieve efficient catalyst design. In addition, the data in this work is detailed and convincing, and the entire manuscript is also well written. I highly recommend publishing this interesting, novel and valuable work, only minor revision is needed:

1 Unit should be given in "TSBC: 32.8 (L1), 30.3 (SpiroBpy), 32.1 (tmphen)"

2 Thermodynamic data is dependent of temperature, so the temperature used for calculating free energy and enthalpy should be mentioned in main text or caption of Fig. 3.

3 Boron and carbon share the same color (cyan) in Fig. 3b, which confuses readers. Boron should be rendered by another color.

Reviewer #3 (Remarks to the Author):

The manuscript outlines the acceleration of Ir-catalyzed arene C-H borylation reaction using a spiro-bpy ligand. The work demonstrates that the TS of the rate-limiting C-H oxidative addition step can be stabilized by the non-covalent interaction between the CH from the spiro-ligand backbone and the arene moiety of substrates. The results are fully supported by theoretical calculations comparing the TSs derived from spiro-bpy with those bearing other related bpy-type ligands, along with interesting KIE analysis by comparing the reaction rate with spiro-bpy and its d8 analog. This work would be of interest to a broad audience, particularly from the synthetic and organometallic communities, in terms of its potential extension to novel ligand designs. Therefore, I am inclined to support its publication in Nat comm, contingent upon the following concerns being adequately addressed.

1. Can the CH-pi interaction also affect the rate of the CB-forming reductive elimination?

2. The reviewer (and readers too) is interested in the effect of the directionality of the CH of spiro-bpy. In this regards, L1 ligands bearing Ph or alkyl groups at the CH2 position of the ligand may provide some insights.

3. The claim that “the reaction using SpiroBpy is enthalpically favored, while the one using SpiroBpy-d8 is entropically favored, in agreement with the stronger KIE at higher temperature and a previous report.” should be explained in more details. Even if it can be preliminary, it should be assumed that some readers do not understand its relationship with the observed rate-acceleration by spiro-bpy-d8.

Response to the Comments of Reviewer 1

In this study, the authors present an excellent strategy for the iridium-catalyzed C–H borylation of electron-rich di-substituted arenes, employing a nonconventional spirobipyridine ligand. This ligand, akin to their previously utilized ligand for undirected meta-selective C–H activation of arenes through remote steric control, exhibits significantly enhanced reactivity compared to conventionally used dtbpy and tmphen ligands. The authors attribute this heightened reactivity to the involvement of a CH– π interaction between the C–H bond of the ligand backbone and the π -electrons of the arene substrate, as proposed through computational studies and ligand kinetic isotope effect analysis. The comprehensive substrate scope exploration includes electron-rich di-substituted arenes, demonstrating high isolated yields and broad functional group tolerance. Additionally, the borylation of few medicinally relevant arenes is showcased.

The authors' assertion about the pivotal role of CH– π interaction in the increased reactivity of the SpiroBpy ligand is intriguing. I share the authors' vision that the thoughtful exploration of such weak noncovalent interactions between ligands and substrates holds promise for advancing ligand design in transition metal catalysis.

We thank the reviewer for the positive evaluation.

While the work is compelling, I have some queries and suggestions that I believe could enhance the clarity and impact of the study. Following necessary revisions, this work has the potential to make a valuable contribution to Nature Communications and capture the attention of the synthetic chemistry community.

1. It is requested to the authors to take into account the acidity of the ortho proton in the ligand participating in the CH– π interaction with the substrate. To improve conformation, it may be useful to introduce an electron-withdrawing substitution adjacent to that proton, rather than Ph or Bpin, to increase the acidity of the proton and enhance its participation in the CH– π interaction.

We thank the reviewer for this important suggestion. Synthetically speaking, we can place substituents at the “meta” position of the fluorenone backbone, as in entries 4–6 in Table 1. However, these substituents also sterically affect the approach of the substrate (see our previous work dedicated to this subject, Ref. 20), and mask the effect of the weak CH– π interaction. To properly answer reviewer's question, we should introduce electron-withdrawing groups further away in the fluorenone backbone, but unfortunately those sites are synthetically difficult to access. We hope to develop a synthetic method for synthesizing such ligands, which preliminary calculations suggest may indeed be more active.

2. In the introduction part an important review for the use of noncovalent interaction is missing (Tetrahedron Letters 59, 1269-1277). It is suggested to cite this reference.

We apologize for this omission, the review was added as new reference 7.

3. To gain better insights into whether the π electron cloud of the substrate is indeed involved in non-covalent interaction with the ligand's ortho proton, it would be nice to check a ligand with a specific substitution (e.g., methyl group) at that position. This alteration aims to provide clearer clarification.

Replacing the H responsible for the CH- π interaction with a methyl group would not only cut off this interaction, but also sterically hinder the approach of the arene substrate, complicating the discussion. This is precisely why we chose D to replace this H, and we observed a clear effect on the reaction rate, supporting the involvement of the C-H bond in the ligand backbone into the energetics of the TS.

4. In the KH/KD experiments, the authors replaced all protons in the ligand with deuterium, creating uncertainty about which specific proton of the ligand is engaged in the proposed non-covalent interaction. In my opinion, the authors might enhance clarity by introducing deuterium solely at the ortho position of the ligand and conducting the control experiment accordingly.

Please note that in order to minimize the potential influence of other C-H bonds, we replaced only the hydrogens of the ligand backbone with deuterium (i.e., no D atoms on bipyridine). We agree with the reviewer that a ligand having only the "ortho" H selectively replaced with D would be the best for the KIE experiments, but unfortunately selective D incorporation at that position is very difficult.

We also wish to point out that by inspecting the structure of the TS and IGMH analysis (Fig. 3b), the other hydrogens from the ligand backbone seem too far to interact significantly with the arene in the TS.

5. Given the significance of the π cloud in substrate interactions, the authors opted for di-substituted arenes with electron-donating properties. However, for a comprehensive comparison and to validate the proposed hypothesis, it would be nice if the authors may examine the reaction with electron-deficient arenes.

Please note that we already investigated such substrates (**2x** in Fig. 2). The SpiroBpy ligand still gave the best yield, but such electron-deficient substrates also reacted well in the presence of the standard dtbpy ligand. It seems that the acceleration effect is larger for more electron-rich arenes, which is consistent with an NBO analysis showing donor-acceptor interactions where π orbitals of the arene substrate is a donor and the C-H σ^* orbital of the ligand backbone is an acceptor.

6. If possible, it would be beneficial to present examples involving mono-substituted arenes,

encompassing both electron-rich and electron-deficient substrates such as anisole, N,N-dimethyl aniline, tBu-benzene, trifluoromethyl benzene, among others.

We mainly investigated disubstituted arenes in order to avoid the extra complications from regioselectivity issues and diborylation, as we also explained in the text: “For convenience in studying the efficiency of the reaction without interference from regioselectivity issues, we chose *meta*- or *ortho*-disubstituted, and polysubstituted arenes as the substrate.”

Just to make sure the reactivity of monosubstituted arenes parallels that of the disubstituted ones, we conducted the reaction of two monosubstituted electron-rich arenes, as shown below. Unsurprisingly, in the same manner with disubstituted arenes, SpiroBpy gave the best overall yields (mixture of regioisomers and diborylated products).

7. Finally, it is suggested that the authors include citations to other pertinent research papers and review articles addressing non-covalent interactions and catalyst engineering. (Such as: Chem. Soc. Rev., 51, 5042–5100 (2022), J. Am. Chem. Soc., 143, 5022–5037 (2021), Science Advances 2023, 9, eadg3311.).

We added these articles as new references 31, 47, and 53, respectively.

Collectively, I have enjoyed a lot to read this paper and I strongly believe that this contribution would certainly be impactful for the catalysis research and for all types of synthetic organic chemistries.

We thank the reviewer for the kind words, and we hope that our work will benefit the synthetic community and promote new avenues for ligand design based on weak noncovalent interactions.

Response to the Comments of Reviewer 2

The author proposed a novel method to improve the efficiency of Ir complex-catalyzed C-H borylation of arenes, that is, through the appropriate selection of Ir ligands, stabilization of transition state of the undirected aromatic C-H bond cleavage step is achieved by forming C-H... π non-covalent interaction between the ligand and the substrate. The author proved the effectiveness of this idea through comprehensive experiments and screened out the most ideal ligand, spirobipyridine. The article further demonstrates from a theoretical perspective that C-H... π interaction improves catalytic efficiency through rigorous DFT calculations, and clearly demonstrates the existence of C-H... π interaction through the very intuitive IGMH analysis. This is an outstanding work that combines theory and experiment, and it is also a very successful example of utilization of weak interactions to achieve efficient catalyst design. In addition, the data in this work is detailed and convincing, and the entire manuscript is also well written. I highly recommend publishing this interesting, novel and valuable work, only minor revision is needed:

We thank the reviewer for the positive evaluation.

1 Unit should be given in "TSBC: 32.8 (L1), 30.3 (SpiroBpy), 32.1 (tmphen)"

We added the unit (kcal mol^{-1}) to all energies in the text.

2 Thermodynamic data is dependent of temperature, so the temperature used for calculating free energy and enthalpy should be mentioned in main text or caption of Fig. 3.

The temperature (25 °C / 298.15 K) was mentioned in the caption of Figure 3a in addition to the supplementary information.

3 Boron and carbon share the same color (cyan) in Fig. 3b, which confuses readers. Boron should be rendered by another color.

The color of the boron atoms was changed to pink.

Response to the Comments of Reviewer 3

The manuscript outlines the acceleration of Ir-catalyzed arene C-H borylation reaction using a spiro-bpy ligand. The work demonstrates that the TS of the rate-limiting C-H oxidative addition step can be stabilized by the non-covalent interaction between the CH from the spiro-ligand backbone and the arene moiety of substrates. The results are fully supported by theoretical calculations comparing the TSs derived from spiro-bpy with those bearing other related bpy-type ligands, along with interesting KIE analysis by comparing the reaction rate with spiro-bpy and its d8 analog. This work would be of interest to a broad audience, particularly from the synthetic and organometallic communities, in terms of its potential extension to novel ligand designs. Therefore, I am inclined to support its publication in Nat comm, contingent upon the following concerns being adequately addressed.

1. Can the CH- π interaction also affect the rate of the CB-forming reductive elimination?

This is a certain possibility that we cannot exclude at the moment.

Previous knowledge for iridium/bipyridine-catalyzed borylation of arenes suggests that the oxidative addition is the turnover-limiting step, and therefore we expect the CH- π stabilizing effect to be the strongest for the TS of the oxidative addition step. A preliminary computation of the TS for the reductive elimination step indicated that the energy of this step is ca. 5.8 kcal/mol lower in Gibbs free energy than that of the oxidative addition step, consistent with oxidative addition being the turnover-limiting step, and with a general trend that the oxidative addition is slower and reductive elimination is faster with more electron-rich arenes.

2. The reviewer (and readers too) is interested in the effect of the directionality of the CH of spiro-bpy. In this regards, L1 ligands bearing Ph or alkyl groups at the CH₂ position of the ligand may provide some insights.

We conducted the reaction with a newly synthesized ligand bearing two Ph groups (**L2**, Table 1, new entry 8), to obtain the product in 65% yield. This result suggests that a certain degree of noncovalent interaction is involved, but not as significant as for the rigid SpiroBipyridine ligand.

3. The claim that “the reaction using SpiroBpy is enthalpically favored, while the one using SpiroBpy-d8 is entropically favored, in agreement with the stronger KIE at higher temperature and a previous report.” should be explained in more details. Even if it can be preliminary, it should be assumed that some readers do not understand its relationship with the observed rate-acceleration by spiro-bpy-d8.

We added one more explanatory (and speculative) sentence, based on the discussion at Ref. 65:

“Although speculative at the moment, this could be partially explained by the stronger CH- π interaction and shorter CH- π distance, which are a result of the longer C-H bond than the C-D bond and the slightly larger polarizability of hydrogen than that of deuterium.”

REVIEWERS' COMMENTS

Reviewer #1 (Remarks to the Author):

I am pleased to see the revised version. I fully understand the difficulty to address some of the important points raised by this reviewer. I also believe that the authors will publish a follow-up paper disclosing all the details as stated by the authors. The author has given enough efforts to improve the manuscript. Considering the pioneering reports in this area of research and huge importance of this present work, I strongly support for its publication in Nature Communications without any revision.

Reviewer #2 (Remarks to the Author):

The authors have appropriately revised this manuscript based on my comments. This manuscript now is fine enough and can be accepted without any other modifications.

Reviewer #3 (Remarks to the Author):

The revised manuscript has addressed most of the important issues raised by the previous reviewers and can be accepted in its current form.